# Seasonal change of *Burkholderia pseudomallei* in paddy field water strongly correlates with ambient temperature: A study in north-central Vietnam

Thi Ngoc Anh Vu, Thi Le Quyen Tran, Nguyen Hai Linh Bui, Trung Thanh Trinh ⓘ*

VNU-Institute of Microbiology and Biotechnology, Vietnam National University, Hanoi, Vietnam

* tttrung@vnu.edu.vn

## Abstract

Melioidosis, a fatal infectious disease caused by the environmental bacterium *Burkholderia pseudomallei*, is known to be associated with the rainy season. Although several attempts have been made to explain this phenomenon, data on the association between the presence of *B. pseudomallei* in environmental reservoirs and weather variables remain limited. This study focused on water samples collected from a paddy field in north-central Vietnam to investigate this association. A total of 800 samples were collected over eight different time points in 2018. Using a two-step enrichment method, 349 samples (43.6%) were positive by a *B. pseudomallei*-specific real-time PCR assay targeting the TTSS1 gene and by *B. pseudomallei* isolation on Ashdown agar. The positive culture rate of *B. pseudomallei* ranged from 5% in the winter to 82% in the summer. Quantitative culture method directly detected *B. pseudomallei* colonies only from samples collected in the summer, with an overall mean count of 93.1 CFU/ml (n = 13; range from 5 to 750). The positive culture rate of *B. pseudomallei* showed a strong positive correlation with the average ambient temperature when binned from three days up to a month before the sampling dates, with the strongest correlation observed at the 19-day bin data ($r_s$ = 0.905; p = 0.002). Two clusters of melioidosis cases were identified in the summer, one after tropical depressions and another after rice harvesting activities. Diverse *B. pseudomallei* genotypes were found within this small-scale paddy field, with a novel sequence type (ST) 1994 persisting throughout the year. In conclusion, our study demonstrated that increased ambient temperature significantly contributed to the higher occurrence and bacterial load of *B. pseudomallei* in surface water, leading to more melioidosis cases if combined with severe rainfalls or occupational agricultural exposure. These findings should be considered in the context of global warming and climate change.

**Data availability statement:** The authors confirm that all data underlying the findings are fully available without restriction. All relevant data are within the paper and its Supporting Information files.

**Funding:** The author(s) received no specific funding for this work.

**Competing interests:** The authors have declared that no competing interests exist.

## Author summary

*Burkholderia pseudomallei* is an environmental bacterium that can cause melioidosis, a potentially fatal infectious disease affecting both humans and animals, particularly in many tropical and subtropical regions. While melioidosis cases typically peak during the rainy season, the association between the presence of this bacterium in environmental reservoirs, climatic factors, and disease incidence remains poorly understood. This study provides the first evidence of a significant correlation between ambient temperature and the positive culture rate of *B. pseudomallei* in paddy field water. The findings suggest that the increase in melioidosis cases during the rainy season can be attributed to a higher occurrence and bacterial load of *B. pseudomallei* in surface water during the summer, especially after heavy rainfalls and occupational agricultural exposure. Our findings are important to consider in the context of global warming and climate change, which may shift the global distribution of *B. pseudomallei* and the epidemiology of melioidosis, possibly expanding into areas in subtropical regions where melioidosis is currently suspected to be less common.

## Introduction

The environmental bacterium *Burkholderia pseudomallei* is a causative agent of melioidosis, a potentially fatal infectious disease in humans and a wide variety of captive and wild animals in the tropics and subtropics [1,2]. In highly endemic areas, the bacterium can be found at a relatively high cell count in soils and surface waters [3–5]. Human infections may occur via percutaneous inoculation of the bacterium when exposed directly to the contaminated soils and surface waters during daily working activities, inhalation of the aerosolized bacteria generated during extreme weather events such as tropical cyclones or typhoons, or ingestion of untreated drinking water and uncooked foods [6–8]. Thus, farmers who regularly work in agricultural fields and live under poor sanitation and hygiene conditions in rural areas are at the highest risk of infection. Diagnosis of the disease is very challenging, particularly in resource-limited areas of many developing countries where the burden of melioidosis is highly suspected [9,10]. A practical approach to detecting melioidosis cases involves raising awareness of the disease in clinical settings and improving culture detection of the bacterium from clinical specimens [11]. Additionally, environmental surveillance of *B. pseudomallei* in soils or waters offers an alternative approach to identifying any potential areas at risk of melioidosis [12,13].

Reports from many endemic regions have shown that melioidosis occurs periodically, with a significant increase in the number of cases detected during the rainy season. Several attempts have been made to explain this phenomenon, including analyses of the association between disease incidence and weather variables, the seasonal presence of *B. pseudomallei* in soils, or human activities associated with increased risk of infection [7,8,14]. Among weather variables, heavy rainfall and

strong winds are well-known climatic factors contributing to the increased risk of exposure to *B. pseudomallei*, either by generating bacterium-containing aerosols or by facilitating bacterial spread in runoff water [6,8,15]. Additionally, rainfall replenishes soil moisture content, which significantly impacts the presence and bacterial load of *B. pseudomallei* in the superficial soils during the rainy season [8,14]. Other weather variables, such as humidity, cloud cover, and groundwater levels, have also been reported to contribute to disease incidence in both dry and wet tropics [16–18]. However, the role of ambient temperature in this phenomenon remains unclear, although a rise in sea surface temperature and maximum ambient temperature has been associated with increased melioidosis cases in northern Australia [17].

The presence of *B. pseudomallei* in environmental reservoirs has been reported to be associated with various factors, including weather variables, soil physicochemical properties, water parameters, biological factors, and human agricultural practices [5,14,19,20]. Both laboratory and field investigations have demonstrated that the growth and persistence of *B. pseudomallei* in soils depend significantly on moisture content [8,14,21]. This presents challenges for studying the association between the presence of *B. pseudomallei* in the superficial soil layer and other weather variables, particularly in regions with distinct dry and rainy seasons. To address this issue, our study focused on collecting water samples from a waterlogged paddy field to examine the presence of *B. pseudomallei* throughout different time points of the year. Then, we analyzed the correlations between the positive culture rate of *B. pseudomallei* in the paddy field water and weather variables. Additionally, we discuss how the seasonal presence of *B. pseudomallei* in the paddy field water, together with weather events and agricultural activities, contributes to the increase in melioidosis cases.

## Materials and methods

### Ethics statement

Ethical approval was not required for the collection of soil and paddy field water samples. The study also included retrospective work involving the collection of clinical *B. pseudomallei* strains and basic demographic information from patients with culture-confirmed melioidosis diagnosed at provincial hospitals. Therefore, written informed consent was not applicable for the retrospective investigation. The protocols for collecting the bacterial strains and clinical information of the patients were approved by the Ethical Committee in Biomedical Research, University of Medicine and Pharmacy, Vietnam National University, Hanoi (Approval No. 04/2020/CN-HDDD).

### Study site and environmental sampling

The study site was located in north-central Vietnam, a region that lies between two classified tropical monsoon climate (Am) and humid subtropical climate (Cwa) types [22], with four distinct winter, spring, summer, and autumn seasons. Winter typically lasts from December to February, spring from March to May, summer from June to August, and autumn from September to November. The rainy season covers from late spring to autumn, and the weather is influenced by the northeast monsoon, which causes cold and dry climate conditions in early winter and cold and highly humid conditions in late winter [23]. The majority of the local population are farmers, and the regional economy mainly relies on rice cultivation, with two main crops called the winter-spring season and the summer-autumn season. Irrigation and plowing typically begin before each planting season, and paddy fields always remain flooded during rice growth.

During the rainy season in 2015, we conducted a pilot study on the presence of *B. pseudomallei* in paddy soils in this north-central part of Vietnam. Soil sampling was carried out at 37 sites across 3 provinces of Thanh Hoa, Nghe An, and Ha Tinh (S1 Fig). At each sampling site, five soil samples were collected, with each sample taken approximately 10 meters apart. Upon arrival at the laboratory after transport at ambient temperature, the soil samples were processed within two weeks for the qualitative culture of *B. pseudomallei*, as described below. Only two sites were positive for *B. pseudomallei*. One site in Can Loc district, Ha Tinh province, had one out of five soil samples positive, while another site (located at coordinates 19º50'18.8" N; 105º37'13.4" E) in Trieu Son district, Thanh Hoa province, had all five soil samples

positive for *B. pseudomallei*. At the time of visiting this latter site on May 29[th], 2015, the rice had been harvested, and the field was drained of surface water. Based on the pilot study results, we selected this paddy site for further investigation.

In 2018, we revisited this rice paddy site to collect water samples throughout the year. On the first visit, we walked along the edge paths of the rice field and marked 50 sampling points with bamboo stakes (S2 Fig). The distance between two consecutive sampling points was approximately 15 m (Fig 1B). At each point, two flooded water samples were collected, one from the right-hand path and another from the left-hand path, using new 500 ml commercial plastic bottles commonly used for bottling drinking water. New disposable gloves were worn for each sample. Each bottle was opened and carefully submerged just below the surface of the flooded water to avoid disturbing the underlying soil layer (S2 Fig), and at least two-thirds of the bottle's volume was filled with paddy field water. After tightly closing the bottle, its exterior was sprayed with 70% alcohol for disinfection before labeling. The distance between right and left samples at each point was approximately 1 m. At each visit, a total of 100 water samples were collected across the rice field, covering an area of approximately 110,000 m$^2$ (Fig 1B). Additionally, between 5 and 24 water samples were collected from the adjacent irrigation canal during each visit (Fig 1B). Water samples were transported to the laboratory on the same day at ambient temperature and were processed for qualitative bacterial culture within 3 days of arrival. Then, the samples were stored at room temperature for subsequent quantitative bacterial culture, as described below. Over the year, eight sampling visits were conducted on February 24[th], April 8[th], May 7[th], May 21[st], June 10[th], July 15[th], August 14[th], and September 19[th]. These visits covered two rice crops: the first winter-spring one from the middle of February to the end of May and the second summer-autumn one from the beginning of July to the middle of October (Fig 2).

## Qualitative and quantitative culture for *B. pseudomallei* and other bacteria

For qualitative culture of *B. pseudomallei* from soil samples collected in 2015, isolation procedure was performed by a conventional method using Galimand's broth for the selective enrichment step and Ashdown agar for the plating-out step, as previously described by Smith et al. (1995) [3].

For the qualitative culture of *B. pseudomallei* from paddy field water samples collected in 2018, an isolation procedure using a two-step enrichment method was performed as previously described [24,25]. Briefly, 100 ml of each water sample was centrifuged at 5,000 rpm for 30 min using a 50 ml Falcon tube (centrifuged twice). Then, the supernatant was decanted to collect the water sediment. After adding 20 ml of TBSS-C50 broth, the tube was vortexed vigorously before being incubated statically at 40°C for 2 days. Subsequently, 1 ml of the first culture supernatant was transferred to a new 50 ml tube containing 9 ml of erythritol medium (EM). The EM medium was developed by replacing the L-threonine and nitrilotriacetic acid in the TBSS-C50 medium with erythritol and $NH_4H_2PO_4$ to facilitate the growth of *B. pseudomallei,* but not other closely related species such as *B. thailandensis* [24]. The tube was incubated at 40°C for 4 days. Afterward, 500 μl of the second culture supernatant was transferred to a new 1.5 ml tube and centrifuged at 8,000 rpm for 10 min to obtain a bacterial cell pellet. Following DNA extraction using the chloroform: isoamyl alcohol method, *B. pseudomallei* was detected by a specific real-time PCR assay targeting the TTSS1 gene [24]. To confirm the presence of *B. pseudomallei*, the second culture supernatant, positive by the specific real-time PCR assay, was plated onto Ashdown agar. After incubation at 40°C for 4 days, suspected *B. pseudomallei* colonies (described below) were picked up, identified by the specific real-time PCR assay, and stored at -70°C in Luria-Bertani broth supplemented with 20% glycerol. A positive culture of *B. pseudomallei* was defined by both the detection of the *B. pseudomallei*-specific TTSS1 gene and the successful isolation of *B. pseudomallei* strains from the second enriched culture supernatant.

For the quantitative culture of *B. pseudomallei* and other bacteria from paddy field water samples, 200 μl of water sample that tested positive by the specific real-time PCR assay was directly plated onto Ashdown agar plates. After incubation at 40°C for 4 days, the suspected *B. pseudomallei* and other bacterial colonies were counted, and the number of colony-forming units (CFUs) per 1 ml of paddy field water was calculated. Colonies suspected to be *B. pseudomallei* typically appeared dry and wrinkled surfaces with a dark violet color and were confirmed positive by the *B. pseudomallei*-specific

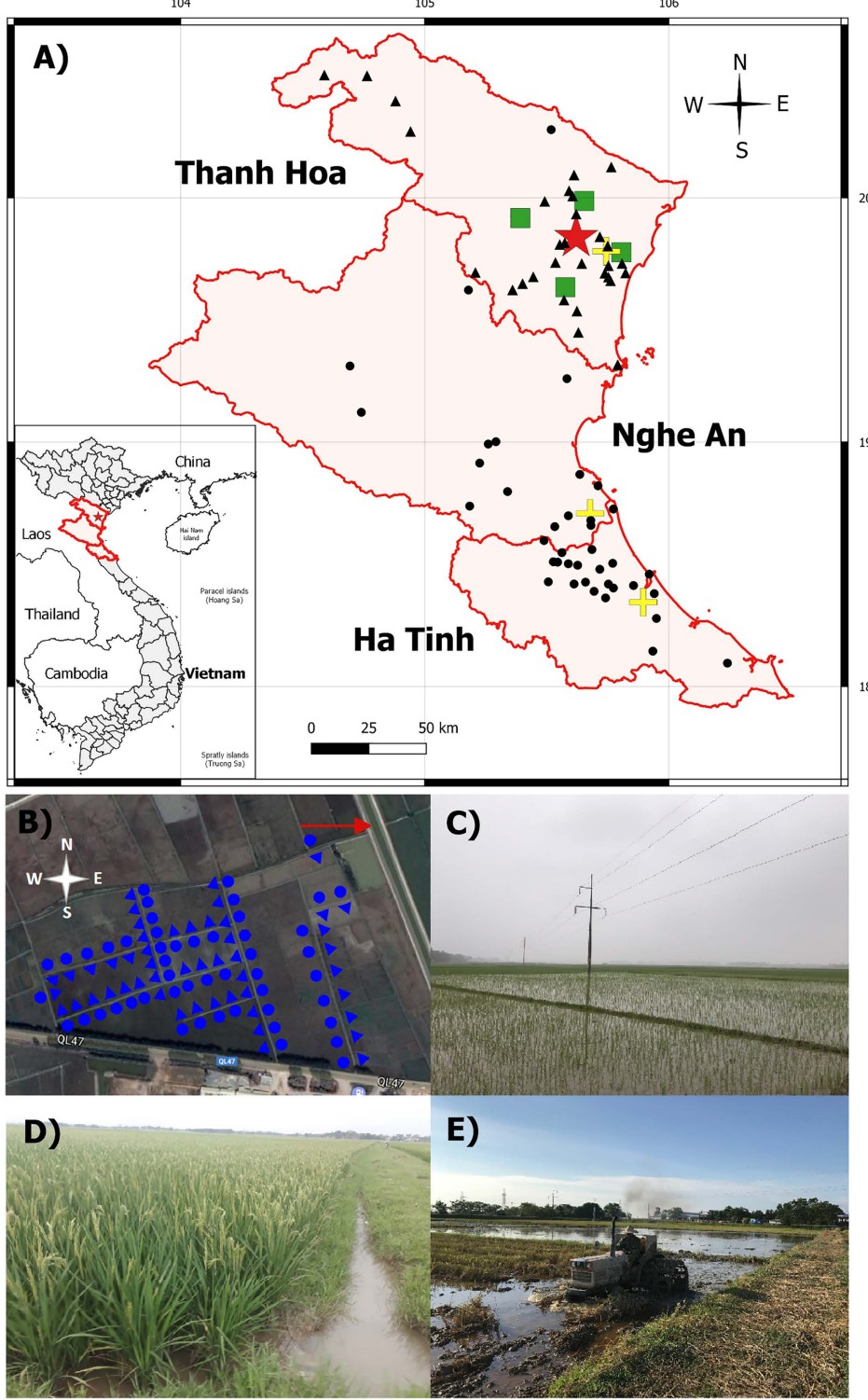

**Fig 1. Study location. (A)** Map of Thanh Hoa, Nghe An, and Ha Tinh provinces in north-central Vietnam. The red star is the sampling site for collecting paddy field water samples. The green squares are the locations of four nearby weather stations serving for meteorological data collection. The yellow plus symbols indicate locations of three provincial hospitals of Thanh Hoa, Nghe An and Ha Tinh, where information on melioidosis cases was collected. The black circles represent the distribution of culture-confirmed melioidosis cases based on the home addresses of patients admitted to the provincial

hospitals of Nghe An and Ha Tinh in 2018, whereas the black triangles indicate melioidosis patients admitted to the provincial hospital of Thanh Hoa in 2019 and 2020. **(B)** The blue circles and triangles represent the water samples collected from the left-hand path and right-hand path, respectively, at each sampling point in the rice field. The red arrow shows the adjacent irrigation agriculture canal. **(C)** The rice field at the first winter-spring cultivation on February 24th. **(D)** Rice plants on May 8th. **(E)** Plowing activity after harvesting the first winter-spring rice crop on June 10th. The map was created using the QGIS software version 3.22.1. The basemap shapefile was downloaded from the Database of Global Administrative Areas https://gadm.org/download_country.html.

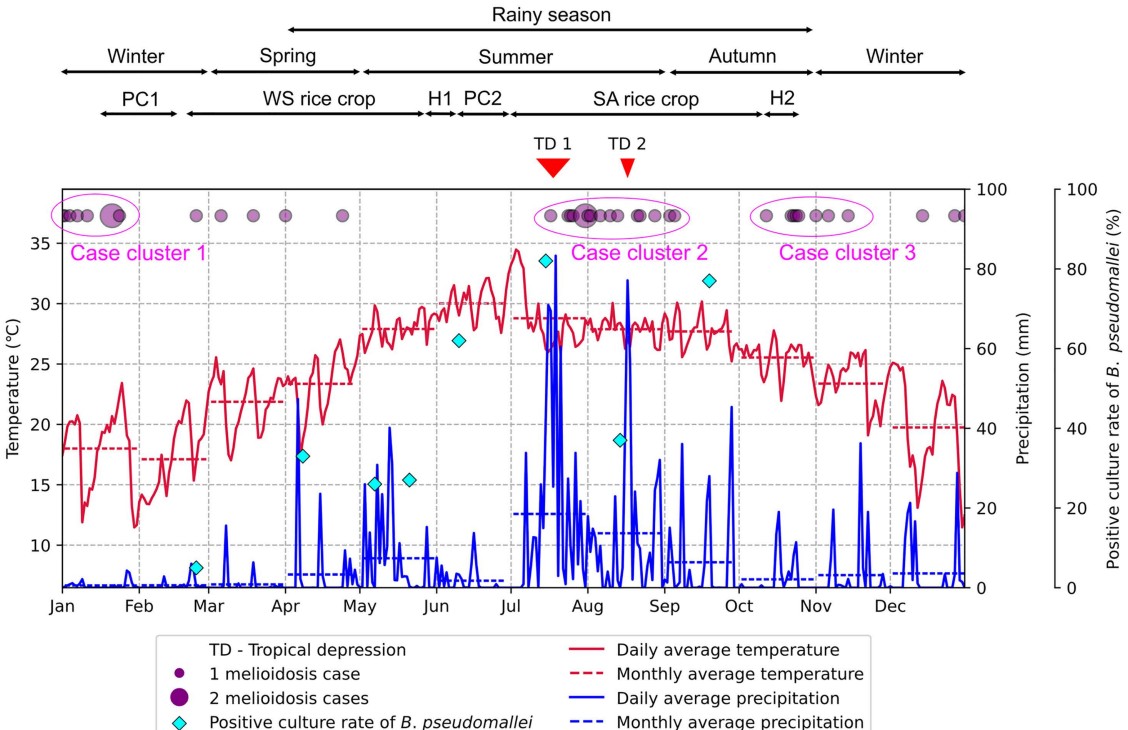

**Fig 2. Daily average temperature, total precipitation, positive culture rate of *B. pseudomallei*, and melioidosis cases occurred in 2018.** The red and blue lines represent the daily average temperature and total precipitation, respectively. The dashed bars show the monthly average weather variables. The cyan diamonds indicate the positive culture rate of *B. pseudomallei* at different sampling dates. The pink circles represent the dates when melioidosis patients were admitted to the hospitals, with the size of each circle proportional to the number of cases admitted on that day. The pink ovals are clusters of melioidosis cases observed. The red triangles indicate the occurrence of two tropical depressions (TD1 and TD2) that affected north-central Vietnam in 2018. The upper arrows indicate different seasons and the order of agriculture activities throughout the year, including plowing and cultivation (PC1) and harvesting (H1) in the first winter-spring (WS) rice crop, and plowing and cultivation (PC2) and harvesting (H2) in the second winter-spring (WS) rice crop.

real-time PCR assay. In contrast, other bacterial colonies, suspected to be some common species growing on Ashdown agar, such as *B. cepacia* complex, *B. thailandensis*, *Pseudomonas* spp., or *Ralstonia* spp, exhibited smooth and mucoid appearances. These other bacterial species were confirmed negative by the *B. pseudomallei*-specific real-time PCR assay.

## Weather data collection

The daily weather data at four nearby weather stations of Bai Thuong, Thanh Hoa, Yen Dinh, and Nhu Xuan were obtained from the Vietnam Meteorological and Hydrological Administration. The data, summarized in S1 Table, included

records of minimum, maximum, and average temperature (°C), total precipitation (mm), minimum and average humidity (%), and wind speed at four different time points throughout the day (m/s).

### Melioidosis case data collection

Following a series of clinical awareness raising on melioidosis and microbiological hands-on training for the detection of *B. pseudomallei* from clinical specimens [11], we reviewed the microbiological laboratory logbooks to collect demographic information on patients with culture-confirmed melioidosis diagnosed at the provincial general hospitals of Nghe An and Ha Tinh in 2018. Patients who were readmitted within a year due to relapse or reinfection were excluded. Based on the patient's home addresses, the geographical distribution of the melioidosis cases was mapped using the QGIS software version 3.22.1 (Fig 1A). To confirm the presence of melioidosis in Thanh Hoa province, we also reviewed the microbiological laboratory logbooks at the provincial general hospitals of Thanh Hoa in 2019 and 2020 and mapped the culture-confirmed cases with melioidosis (Fig 1A).

### MLST genotyping

Multilocus sequence typing (MLST) of *B. pseudomallei* isolates was performed following the previously described protocol [26], with the PCR primers available at the pubMLST database (https://pubmlst.org/organisms/burkholderia-pseudomallei/primers). A new allelic profile was submitted to the curator for the new sequence type (ST) assignment.

### Statistical analysis

All analyses were conducted in a Python (ver. 3.11.4) environment. Statistical tests were performed using the SciPy library (ver. 1.11.4) with a significance level (alpha) of 0.05. Graphs were generated with the Matplotlib library (ver. 3.7.1) and annotated using Photopea (https://www.photopea.com/). Daily weather variables from four stations were averaged into bins ranging from 1 to 30 days before the sampling date. Spearman's rho analysis was used to examine the correlation between the binned weather data and the positive culture rate of *B. pseudomallei*.

### Results

After a two-step enrichment method, 349 (43.6%) out of 800 water samples collected from the paddy field were positive for *B. pseudomallei* DNA by the specific real-time PCR assay. *B. pseudomallei* strains were successfully isolated from all of the second enriched culture supernatants of these water samples. The positive culture rate of *B. pseudomallei* varied at different sampling visits: 5% in February, 33% in April, 26% on May 7th, 27% on May 21st, 62% in June, 82% in July, 37% in August, and 77% in September (Fig 2). The quantitative culture of the paddy field water directly detected *B. pseudomallei* colonies growing on Ashdown agar only from 13 samples collected during the last three sampling time points in the summer and autumn. The overall mean count was 93.1 CFU/ml (range from 5 to 750). The mean counts of *B. pseudomallei* at each sampling time were 32.5 CFU/ml (n = 4; range from 5 to 85), 10 CFU/ml (n = 2; range from 5 to 15), and 151.4 CFU/ml (n = 7; range from 5 to 750) for the samples collected in July, August, and September, respectively. All of the water samples taken from the adjacent irrigation canal were culture-negative for *B. pseudomallei* (Fig 1B).

Average weather data collected from four nearby stations were plotted (Figs 2 and S5), and maximum cross-correlation between the weather variables and the positive culture rate of *B. pseudomallei* was examined. Preceding average ambient temperatures data showed highly relative constant correlations with the positive culture rate of *B. pseudomallei* when the data binned from 9 days up to a month before the sampling date. The strongest positive correlation was observed at 19-day bin data of the daily average temperature (S3 Fig), with a Spearman's rho correlation coefficient of 0.905 (p = 0.002) (Fig 3A). Surprisingly, the positive culture rate of *B. pseudomallei* showed a strong negative correlation with preceding average humidity per bin width ranging from 9 days to 16 days before the sampling date (S3 Fig), with a Spearman's rho correlation coefficient of -0.881 (p = 0.004) at 12-day bin data (Fig 3B).

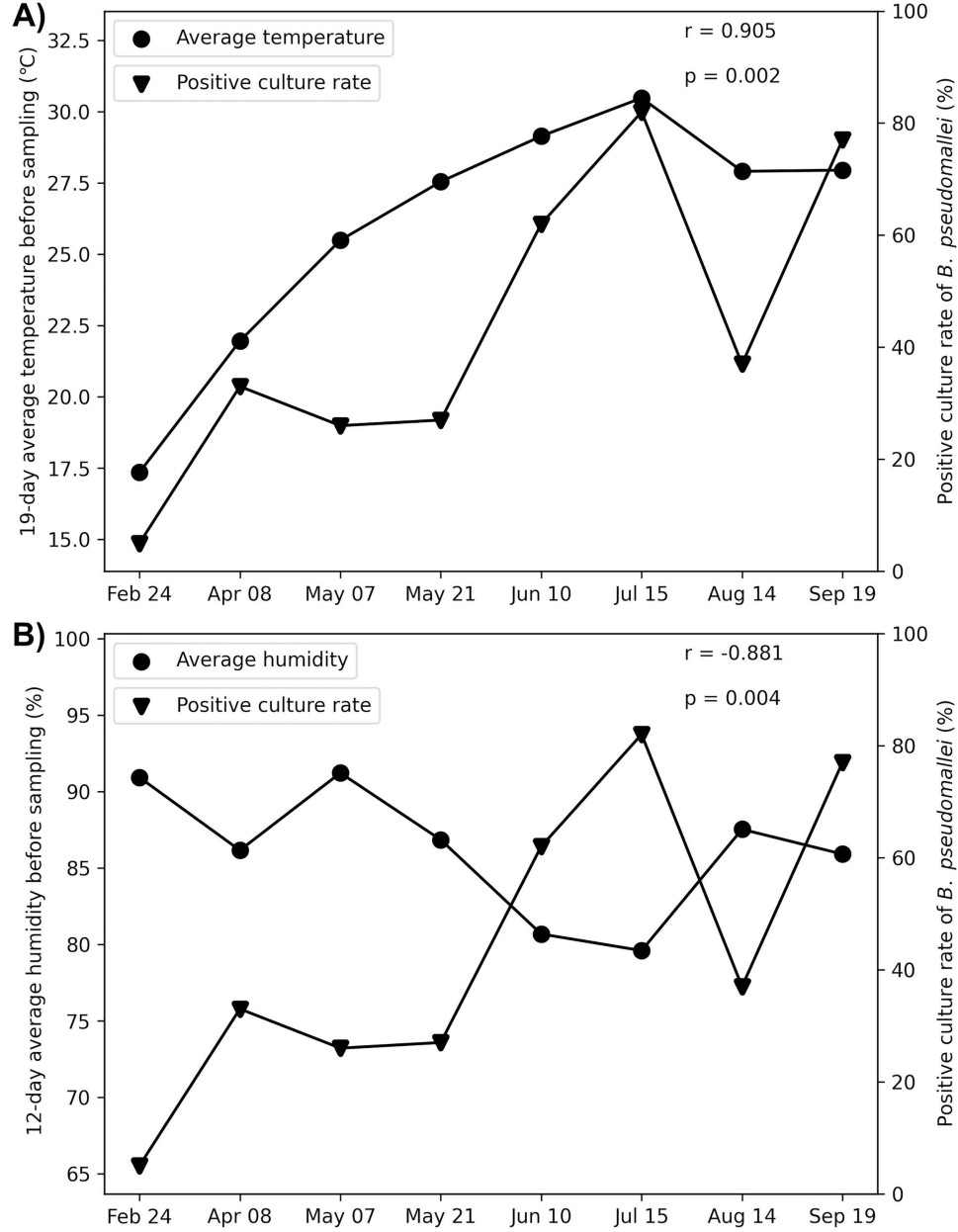

**Fig 3. Correlation between the positive culture rate of *B. pseudomallei* with the ambient average temperature (A) and humidity (B).**

For other bacteria growing on Ashdown agar, we also observed the increase in the mean cell counts associated with the rise of the ambient temperature over the year. The mean cell counts, which ranged from undetectable levels in the winter to 439 CFU/ml in the autumn, were strongly correlated with the 19-day bin data of the daily average temperature, with a Spearman's rho correlation coefficient of 0.810 (p = 0.015) (S4 Fig).

In 2018, 40 melioidosis patients were diagnosed at two nearby provincial hospitals of Nghe An and Ha Tinh. Based on the date of hospital admission, the number of cases were plotted and three clusters of melioidosis cases were detected in January (n = 8), from the middle of July to the beginning of September (n = 16), and from the middle of October to the

middle of November (n = 8) (Fig 2). The second cluster occurred after two tropical depressions affecting the north-central Vietnam on July 18th and August 17th. The last cluster observed after the harvesting season of the second rice crop.

Over the year, the frequency of *B. pseudomallei* isolation at the left-hand and right-hand sides of sampling points ranged from once (e.g., the right-hand sample at sampling point 1, isolated in 1 out of 8 visits) to seven times (e.g., the right-hand sample at sampling point 36, isolated in 7 out of 8 visits) (Fig 4). By mapping the presence of *B. pseudomallei* on both sides of the sampling points during each visit and across the study period, we found that the bacterium was heterogeneously distributed in paddy field water, both spatially and temporally, even within such a small rice field (Fig 4).

Among the *B. pseudomallei* strains isolated from the enriched cultures of soil and water samples collected from the paddy field, we randomly selected 27 strains for the MLST analysis. All seven strains isolated from the soil samples collected in the pilot study had the same ST 543. The remaining 20 strains, isolated from the paddy water samples at different sampling time points, showed greater diversity, including ST 533 (n = 3; 15%), ST 541 (n = 1; 5%), ST 542 (n = 5; 25%), ST 543 (n = 1; 5%), and a novel ST assigned as ST 1994 (n = 10; 50%). The distribution of the water samples positive for *B. pseudomallei* and the corresponding STs of the selected strains are shown in Fig 4.

## Discussion

Human acquisition of *B. pseudomallei* involves multiple factors, including the virulence of the bacterial strains, host susceptibility, transmission routes, and particularly the presence and load of the bacterium in environmental reservoirs [27]. Since *B. pseudomallei* was first recognized as an environmental pathogen, soils have been commonly used as a sample type to detect its presence and identify areas at risk of melioidosis [12,13]. Certain soil characteristics, such as sandy soil type, high moisture content, low pH, low iron level, low nutrient availability, and low organic matter content, have been reported to be associated with the presence of *B. pseudomallei* [14,20]. However, data on the correlation between the seasonal presence of *B. pseudomallei* in soils and annual weather variables remain limited, partly because soil moisture is strongly influenced by occasional rainfall in the wet season [8]. Base on the detection of *B. pseudomallei* in soil samples collected from a rice field in north-central Vietnam, and the typical flooding of paddy fields during plowing, cultivation and crop growth (Fig 1C, 1D, and 1E), we decided to collect water samples from the waterlogged paddy field throughout the year instead of soil samples. This approach aimed to exclude soil moisture as a confounding factor in the correlation analysis. Our data showed that the positive culture rate of *B. pseudomallei* in paddy field water gradually increased when ambient temperatures rose from winter to summer, even before the onset of heavy rains typically beginning in the middle of summer. *B. pseudomallei* load in the paddy field water also increased to detectable levels in the summer.

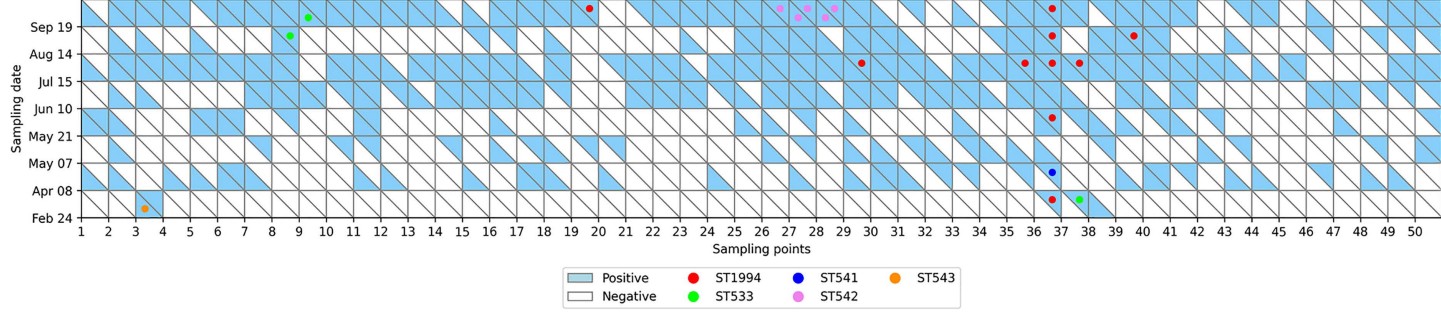

**Fig 4. Distribution map of *B. pseudomallei* in the waterlogged rice field throughout the year.** The upper and lower triangles are water samples collected on the right-hand path and left-hand path at each sampling point, respectively. Blue-colored triangles indicate culture-positive samples for *B. pseudomallei*, whereas non-colored triangles present culture-negative samples. Colored circles correspond to the different sequence types (ST) of *B. pseudomallei*.

Daily minimum, maximum, and average temperatures were all positively correlated with the positive culture rate of *B. pseudomallei* when the data were aggregated into bins ranging from 9 days up to a month, with the strongest correlation observed at a 19-day average temperature. Although temperature and humidity data were not collinear (S1 Table), we observed a strong negative correlation between ambient humidity and the positive culture rate of *B. pseudomallei* in paddy field water, particularly in a short bin width from 9 days to 16 days (S3 Fig). Since ambient humidity is unlikely to affect microbial life in water directly, our findings suggest that ambient temperature was the main climatic factor contributing to the growth and persistence of *B. pseudomallei* in the paddy field water.

Results from laboratory experiments have shown that temperature plays a significant role in the growth and survival of *B. pseudomallei* in spiked soil or water samples. *B. pseudomallei* could survive for over 720 days in normal saline water at temperatures between 24°C and 32°C, but its cultivability decreased significantly when the temperature was below 16°C or above 40°C [28]. In the environment mimicking slurry soils, *B. pseudomallei* exhibited obvious growth at temperatures ranging from 25°C to 42°C (with optimal growth at 37°C and 42°C), while its specific growth rate decreased significantly when temperatures fell below 22°C or rose above 45°C [21]. Our study observed the lowest monthly average temperatures in January and February, with several weeks experiencing daily average temperatures below 16°C. These low temperatures likely affected the growth and survival of *B. pseudomallei*, which may explain the lowest positive culture rates of *B. pseudomallei* observed in the winter. In contrast, the highest monthly average temperature occurred in June, ranging from 27.5°C to 32.5°C, within the temperature range that supported the growth of *B. pseudomallei* [21]. Additionally, an increase to the optimal growth temperature has been reported to promote the expression of flagella in *B. pseudomallei*, potentially facilitating bacterial motility and spread [29]. This combination may explain the highest positive culture rates of *B. pseudomallei* observed during the hot summer and autumn months (Fig 2).

The incubation period for melioidosis has been estimated to be between 1 and 21 days [30]. Recent studies found that the time interval between presumptive infection and hospital admission is within a week after severe rainfalls [16,17] or up to three weeks after normal rainfalls [17]. Among three clusters of melioidosis cases observed in our study, the second cluster occurred after two mild tropical depressions that affected north-central Vietnam: one on July 15th, with a total rainfall of 355 mm over 7 days, and another on August 16th, with a total rainfall of 173 mm over 3 days. This phenomenon was similar to reports from other endemic areas showing increased case numbers after consecutive heavy rainfalls. Notably, the third cluster occurred after rice harvesting activities rather than extreme weather events. There were two rice harvesting seasons in the year: one was at the end of May (the first rice crop), and another was at the end of September (the second rice crop). Although weather variables such as average monthly rainfall and maximum wind speed were similar between May and September (Fig 2), the positive culture rate of *B. pseudomallei* and its cell counts were higher in September. This suggests that a higher occurrence and/or bacterial load of *B. pseudomallei* in paddy field water in September may have contributed to successful infections when farmers were exposed to contaminated surface waters during harvesting activities, resulting in increased melioidosis cases.

During the hot summer and autumn months, we observed a higher occurrence of *B. pseudomallei* in the paddy field water in north-central Vietnam than in northeast Thailand [4]. This higher occurrence may reflect the improved sensitivity of *B. pseudomallei* detection when using the two-step culture method in our study [24]. However, the overall quantitative mean counts in our samples were lower than those reported for paddy field water in northeast Thailand (93.1 CFU/ml versus 200 CFU/ml) [4] but higher than those reported for other water types, such as pond, rain, and well water samples, in northeast Thailand (5.1 CFU/ml) [31], as well as roadside water from Castle Hill in northern Australia (14 CFU/ml) [32]. A lower positive culture rate of *B. pseudomallei* in the middle of August may be explained by the consecutive heavy rainfalls starting in the middle of July, which likely diluted *B. pseudomallei* cell densities in the waterlogged rice field. The rainfalls not only created more ecological niches of surface water that support *B. pseudomallei* growth but also facilitated its dissemination from rice fields into drainage networks, even in the absence of soil erosion, thereby increasing the risk of infection during the rainy season.

Similar to reports from water or soil samples in other endemic areas, our study also observed diverse *B. pseudomallei* genotypes in paddy water samples in such a small-scale field, although the number of isolates selected for molecular typing was limited [31–33]. ST 543 was found in both soil and water isolates. Compared to the MLST database (https://pubmlst.org/organisms/burkholderia-pseudomallei) analyzed for *B. pseudomallei* isolates (n = 174) from Vietnam, accessed on January 6th, 2025, there were shared genotypes of ST 533, 541, 542, and 543 between clinical and water isolates, which might support previous findings that water is the central vehicle for transmitting and acquiring the disease [16,32]. A new ST 1994 was likely the predominant genotype persisting in the paddy field water over the study period. The genotype has not yet been found in our clinical or soil isolates. This may suggest a superior environmental adaptation but less virulence of this genotype in the paddy water sample type. Further investigations with a large sample size from distant locations are needed to better understand these observations.

There are several limitations to this study. First, water temperatures were not directly measured in the paddy field at the time of sampling. Second, water parameters were not analyzed during the study period. Thus, the changes in physicochemical parameters and available nutrients, such as total nitrogen, potassium, and phosphorus levels influenced by fertilization practices, rainfalls, or water irrigations, could not be reported in association with the presence of *B. pseudomallei*. Third, there was under-reporting of clinical cases at the general hospital of Thanh Hoa province, where water sampling was conducted in this study. Until the end of 2018, this hospital had not participated in the training program for detecting *B. pseudomallei* from clinical samples [11], even though 33 melioidosis cases were detected in 2019 and 2020 (Fig 1A). Therefore, the reported number of melioidosis cases likely did not reflect the true number of infected cases in this study area in 2018.

In conclusion, our study found that a rise in ambient temperature is a significant climatic factor contributing to the higher occurrence and bacterial load of *B. pseudomallei* in the surface water. This finding is particularly important in the context of climate change and global warming [34,35]. With the rise in global surface temperature, the change in the global precipitation patterns, and more intense extreme weather events, there may be a shift in the global distribution of *B. pseudomallei* and the epidemiology of melioidosis, possibly affecting areas in subtropic regions where melioidosis is being suspected to be less common.

## Supporting information

**S1 Table. Summary of daily weather data from four weather stations in 2018.**
(DOCX)

**S2 Table. Pearson pairwise correlations between weather variables in 2018.**
(DOCX)

**S3 Table. Summary of demographic characteristics of 40 culture-confirmed melioidosis cases in Nghe An and Ha Tinh provinces in 2018.**
(DOCX)

**S1 Fig. Soil sampling sites for the pilot study conducted in 2015 across three provinces of Thanh Hoa, Nghe An, and Ha Tinh.** Blue asterisks indicate sampling sites negative for *B. pseudomallei*, while red asterisks indicate sites positive for *B. pseudomallei*. Five soil samples were collected at each site, and the size of the asterisks is proportional to the number of *B. pseudomallei*-positive soil samples per site. The map was created using the QGIS software version 3.22.1. The basemap shapefile was downloaded from the Database of Global Administrative Areas https://gadm.org/download_country.html
(TIF)

**S2 Fig. Water collection on the rice field.** (A) Marking of sampling points with bamboo stakes during the first visit to the rice paddy field in 2018. (B) Sampling of paddy field water using a 500 ml plastic bottle.
(TIF)

**S3 Fig. Cross-correlation between weather variables and positive culture rate of *B. pseudomallei.***
(TIF)

**S4 Fig. Bacterial quantification and correlation analysis.** (A) *B. pseudomallei* and other bacterial counts in the paddy field water over the year 2018. (B) Spearman's rho correlation between other bacterial counts in the paddy field water and the 19-day bin data of the average ambient temperature.
(TIF)

**S5 Fig. Weather observations, positive culture rate of *B. pseudomallei*, and melioidosis cases occurred in 2018.**
The red, blue, green, and grey lines represent the daily average temperature, total precipitation, maximum wind speed, and average humidity, respectively. The dashed bars show the monthly average weather variables. The cyan diamonds indicate the positive culture rate of *B. pseudomallei* at different sampling dates. The pink circles represent the dates when melioidosis patients were admitted to the hospitals, with the size of each circle proportional to the number of cases admitted on that day. The pink ovals are clusters of melioidosis cases observed. The red triangles indicate the occurrence of two tropical depressions (TD1 and TD2) that affected north-central Vietnam in 2018. The upper arrows indicate different seasons and the order of agriculture activities throughout the year, including plowing and cultivation (PC1) and harvesting (H1) in the first winter-spring (WS) rice crop, and plowing and cultivation (PC2) and harvesting (H2) in the second winter-spring (WS) rice crop.
(TIF)

## Acknowledgments

We would like to thank staff members at the Department of Medical Microbiology at the general hospitals of Thanh Hoa, Nghe An, and Ha Tinh provinces for providing the bacterial strains and the patient's demographic data.

## Author contributions

**Conceptualization:** Trung Thanh Trinh.

**Data curation:** Thi Ngoc Anh Vu.

**Formal analysis:** Thi Ngoc Anh Vu, Thi Le Quyen Tran.

**Methodology:** Thi Le Quyen Tran, Nguyen Hai Linh Bui.

**Resources:** Trung Thanh Trinh.

**Supervision:** Trung Thanh Trinh.

**Validation:** Nguyen Hai Linh Bui.

**Writing – original draft:** Thi Ngoc Anh Vu, Trung Thanh Trinh.

**Writing – review & editing:** Trung Thanh Trinh.

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
