## [Decision Letter · Decision Letter 0]

9 Mar 2025

PNTD-D-25-00046

Seasonal change of Burkholderia pseudomallei in paddy field water strongly correlates with ambient temperature: A study in north-central Vietnam

Dear Dr. Trinh,

Thank you for submitting your manuscript to PLOS Neglected Tropical Diseases. After careful consideration, we feel that it has merit but does not fully meet PLOS Neglected Tropical Diseases's publication criteria as it currently stands. Therefore, we invite you to submit a revised version of the manuscript that addresses the points raised during the review process.

Please submit your revised manuscript within 60 days May 08 2025 11:59PM. If you will need more time than this to complete your revisions, please reply to this message or contact the journal office at plosntds@plos.org. Please include the following items when submitting your revised manuscript:

We look forward to receiving your revised manuscript.

Kind regards,

Georgios Pappas

Section Editor

Shaden Kamhawi

co-Editor-in-Chief

Paul Brindley

co-Editor-in-Chief

**Journal Requirements:**

1) We noticed that you used the phrase 'data not shown' in the manuscript. We do not allow these references, as the PLOS data access policy requires that all data be either published with the manuscript or made available in a publicly accessible database. Please amend the supplementary material to include the referenced data or remove the references.

2) We have noticed that you have uploaded Supporting Information files, but you have not included a list of legends. Please add a full list of legends for your Supporting Information files after the references list.

3) Some material included in your submission may be copyrighted. According to PLOSu2019s copyright policy, authors who use figures or other material (e.g., graphics, clipart, maps) from another author or copyright holder must demonstrate or obtain permission to publish this material under the Creative Commons Attribution 4.0 International (CC BY 4.0) License used by PLOS journals. Please closely review the details of PLOSu2019s copyright requirements here: PLOS Licenses and Copyright. If you need to request permissions from a copyright holder, you may use PLOS's Copyright Content Permission form.

Potential Copyright Issues:

i) Please confirm (a) that you are the photographer of 1, or (b) provide written permission from the photographer to publish the photo(s) under our CC BY 4.0 license.

ii) Figure 1A. Please (a) provide a direct link to the base layer of the map (i.e., the country or region border shape) and ensure this is also included in the figure legend; and (b) provide a link to the terms of use / license information for the base layer image or shapefile. We cannot publish proprietary or copyrighted maps (e.g. Google Maps, Mapquest) and the terms of use for your map base layer must be compatible with our CC BY 4.0 license.

4) We note that your Data Availability Statement is currently as follows: "All relevant data are within the manuscript and its supporting information files.". Please confirm at this time whether or not your submission contains all raw data required to replicate the results of your study. Authors must share the “minimal data set” for their submission. PLOS defines the minimal data set to consist of the data required to replicate all study findings reported in the article, as well as related metadata and methods (https://journals.plos.org/plosone/s/data-availability#loc-minimal-data-set-definition).

5) Please insert an Ethics Statement at the beginning of your Methods section, under a subheading 'Ethics Statement'. It must include: 

i) The full name(s) of the Institutional Review Board(s) or Ethics Committee(s)

ii) The approval number(s), or a statement that approval was granted by the named board(s)

iii) A statement that formal consent was obtained (must state whether verbal/written) OR the reason consent was not obtained (e.g. anonymity). NOTE: If child participants, the statement must declare that formal consent was obtained from the parent/guardian.]. 

**Reviewers' Comments:**

Reviewer's Responses to Questions

**Key Review Criteria Required for Acceptance?**

**Methods**

-Are the objectives of the study clearly articulated with a clear testable hypothesis stated?

-Is the study design appropriate to address the stated objectives?

-Is the population clearly described and appropriate for the hypothesis being tested?

-Is the sample size sufficient to ensure adequate power to address the hypothesis being tested?

-Were correct statistical analysis used to support conclusions?

-Are there concerns about ethical or regulatory requirements being met?

Reviewer #1: See below

Reviewer #2: Study design appropriate with appropriate methods to address the objectives of the study. Some additional measures could have been included to make findings and conclusions more robust (specifically, water temperature measurements).

Of note, the paragraph lines 223-227 mentions "other bacteria", however the methods related to this aspect are not included in the manuscript; they need to be included in the revision in order to validate the results presented.

Beyond that, sample size and analyses are appropriate and support conclusions.

Reviewer #3: Overall, the methods used to achieve the objectives of the study were suitable and acceptable. Description of the methods however tended to be on the brief side, requiring reader to search and refer to previous studies. Although this is not unusual, making the effort to describe pertinent experimental details would greatly improve readability and help readers to focus more on the current manuscript. Following are questions and comments that naturally surfaced when the manuscript was reviewed:

Regarding Study Site, L127-131:

i) Although it was mentioned that the current study site was selected because it was the same site on which a pilot study was conducted back in 2015 (line 115), the reasons and justifications for conducting the study on this site were not clear. Might it be that this was a convenient sampling, in which the research team by chance managed to convince the paddy field owner to participate in the study? OR The site was nearby to other sites that were previously reported positive for B. pseudomallei?

ii) Is the rice field with the coordinate 19deg50’18.8” N;105deg37’13.4” E (that have all 5 soil samples positive for B. pseudomallei) also the same one sampled for water in 2018?

iii) The description of the water sampling points in the text although is adequate, it could help readers to understand faster if a representative ground photograph showing the sampling points marked with the bamboo stakes can also be included in Figure 1.

Regarding collection of water sample:

L136: It was described that water "...were collected into new 500 ml plastic bottles. Suggestion for clarity and reproducibility of method by other researchers: state the approximate volume of water collected in this study.

i) How was the water collected into the 500 ml plastic bottles?

ii) Was the plastic bottle directly immersed into the water body? OR A water dipper/ladle was used to ladle the water into the plastic bottles?

iii) Were the plastic bottles pre-sterilized type?

Regarding transport of water & soil samples:

i) How were the samples transported after sampling was completed?

ii) How soon were the samples transported to the lab and processed?

Regarding samples processing & handling?

i) How were the water and soil samples processed in the lab prior to plating and molecular analysis?

ii) How were the samples stored before and after processing (e.g. storage temperature & etc.)

Regarding Quantitative Culture for B. pseudomallei:

L166-168: "...500 uL of water sample... was directly plated..."

i) Was the 500 uL water sample pipetted all at once onto the agar plates? OR The 500 uL sample was divided into 2-3 portions, and pipetted onto the agar portion by portion, allowing the agar to dry before pipetting the next portion? This question is asked because an agar plate prepared using the commonly used petri dish (Diameter 90mm, Depth 15mm) can typically absorbed approximately 200 uL of plating liquid at the maximum at one time. To safely plate a volume of 500 uL all at once, it is assumed that larger-sized petri dishes were used in the current study. If this was the case, what was the size of the petri dishes used to plate the 500 uL water sample?

L169: "...other bacterial colonies were counted..."

i) What species or genus constitute the "other bacteria"?

ii) How were colonies of the "other bacteria" differentiated from B. pseudomallei colonies on the agar plate?

**Results**

-Does the analysis presented match the analysis plan?

-Are the results clearly and completely presented?

-Are the figures (Tables, Images) of sufficient quality for clarity?

Reviewer #1: See below

Reviewer #2: Results are clearly presented, with the exception of confusion being created lines 223-227 referring to studies that were not described in methods.

Figures are clear and appropriate, with one exception (Figure 4 requires an axis label for the horizontal axis); narratives describing figures are appropriate and comprehensive

Reviewer #3: In general, the results presented matched the planned analysis techniques and statistical methods. Following are areas where questions and comments surfaced during review of the manuscript:

L225: Is the word "uncountable" in this context meant that there were no bacteria growing on the plate or there were too many bacteria/microorganisms growing on the plate that it was not possible to count the colonies?

Regarding the 40 human melioidosis case data obtained from the hospitals:

If the data was available, it is suggested that the basic demographics of the 40 cases are summarized in a table and included in the result section, information such as patients' age, sex, and occupation. Justification for inclusion: in the absence of this information in the current manuscript, it can only be assumed that all the 40 human cases were rice farmers, which the text in line 233-234 (The last cluster observed after the harvesting season of the second rice crop) seemed to imply.

FIGURE LEGEND:

L494: The dashed bars (showing the monthly average weather variables) took a bit of time to spot in the graphs (due to multiple data being squeezed into the figure).

Line 506: Regarding the phrase "right and left water sample", it is suggested that the description is amended to "...are water samples collected on the right-hand path and left-hand path of each sampling point, respectively", consistent with the text used in the Method section (Line 135)

FIGURES:

Figure 1 (b): It is suggested that a correctly orientated compass point is included in this figure.

Figure 2:

i) Since only ambient temperature demonstrated significant positive correlation with B. pseudomallei positive culture rate, Figure 2 perhaps can be simplified to showcase the ambient temperature data. The rest of the data (humidity, windspeed and precipitation) could be included as Supplementary.

ii) Is "1 melioidosis case" equaled to one confirmed case, and "2 melioidosis case" equaled to two confirmed cases?

Figure 4:

i) It is suggested that the annotation "Sampling points" on the horizontal axis, and "Sampling date" on the vertical axis are included.

ii) Hopefully a high resolution of this figure is available. Also, it is suggested that a lighter shade of blue is used to denote the sites that were positive for B. pseudomallei culture. At the current resolution and color combinations, it was challenging to see which sampling sites had which STs.

**Conclusions**

-Are the conclusions supported by the data presented?

-Are the limitations of analysis clearly described?

-Do the authors discuss how these data can be helpful to advance our understanding of the topic under study?

-Is public health relevance addressed?

Reviewer #1: See below

Reviewer #2: Conclusions are supported by the data; would request limitation be added related to not measuring water temperature at the sampling site at the time of sample collections. Public health relevance is addressed, and study links effectively to clinical findings that were described

Reviewer #3: The conclusions are generally supported by the results presented. The limitations were also mentioned, in particular the awareness that melioidosis cases nearby the study site were not forthcoming. However, it is probably a little less accurate to deduce that melioidosis cases were under-reported in year 2018 at the Thanh Hoa province just because more cases were detected in the subsequent years.

An aspect that was not discussed was why paddy field was selected as the preferred study site, as compared to other type of aquatic environments.

Following are some comments and questions on the Discussion section:

L270: For microorganisms, the term culturability is typically used, e.g. Viable but non-culturable

Line 306: Sounds contradicting. The sentence "groundwater level actually followed the rise of melioidosis cases" is not clear in the context of this paragraph.

Line 302-Line 315: Have difficulty relating the discussion on the effect of ground water level on the bacterial load since the current study did not look at the effect of ground water level.

Line 352: Was it "irritations" or "irrigations"?

**Editorial and Data Presentation Modifications?**

Reviewer #1: See below

Reviewer #2: The following are suggested edits and revisions:

Line 119, correct “spring” to “autumn” prior to “from September to November”

Line 126, consider word change from “water” to “paddy fields” prior to “always remain flooded”

Recommend reorganizing paragraph lines 115-131 to generally describe the region and rice harvest, followed by description of the pilot study (for example, move the first sentence of the paragraph lines 115-116 to line 126 prior to “At each sampling site”, and open the paragraph instead with “The study site lies in north-central Vietnam.”)

Line 130, consider changing “was completely run out of” to “was drained of”

Line 131, consider adding phrase “Based on the pilot study results,” before “this sampling site…”

Consider moving the sentence in lines 104-141 to Line 137 before “At each visit” for better continuity

Line 141, consider changing “It” to “Sampling”

Line 178, consider changing “raising on” to “trainings on”

Paragraph lines 223-227 – this paragraph does not make sense – what “other bacteria” is being referred to? Need to include reference to this in the Materials and Methods, describing culture of “other bacteria”; also, were these bacteria identified? This is very important information to include given that these bacteria were in competition with B pseudomallei in the environment and may have influenced the ability to culture quantitatively

The first sentence of paragraph lines 235-239 is a bit confusing on first read; consider re-wording for greater clarity.

Line 242, consider deleting “requires a complex of” as this confusing wording; consider instead replacing with the word “involves”

Lines 250-253 – need a reference for this sentence

Figure 4 – need to have label for horizontal/x-axis

Reviewer #3: Revision of this manuscript falls in between Minor and Medium Revision.

**Summary and General Comments**

Reviewer #1: PNTD-D-25-00046

"Seasonal change of Burkholderia pseudomallei in paddy field water strongly correlates

with ambient temperature: A study in north-central Vietnam."

Vu, A.T.N., et al.

This paper evaluates the presence of Burkholderia pseudomallei in paddy fields in Vietnam and the impact seasonal changes are having in the isolation of the bacteria in the soil. The data indicates that changes in temperature have a direct impact in the number of bacteria recovered from the sampling sites, which is an important finding for endemic areas. However, the manuscript requires significant editing and interpretation of the data requires further analysis. The authors should strongly consider the following points to improve the paper:

Major points:

1. Unfortunately, the manuscript requires significant attention to the grammar and additional editing of the manuscript is required. Normally I do not consider that an issue, but this manuscript requires attention, so the data presented can be highlighter. Some examples are provided below.

2. Abstract lane 43, is prevalence what is impacted by changes in ambient temperature? Although prevalence is defined as the number of subjects either at a specific point in or over a specified period, the changes in temperature are only affecting the threshold of bacterial recovery because the microorganism is already prevalent in the soil.

3. Labeling of the panels and description in the text needs to be improved. For example, in lanes 138-140, the panels A through e need to be depicted in the text.

4. Fig 2, lanes 141-143 and discussion, it is evident that case clusters 1 and 3 do not match case cluster 2 with respect to average temperature and precipitation. This is an important point of discussion because you might expect, based on the overall premise, that increase in temperature also produce an increase in bacterial recovery and perhaps cases. However, the data does not support this trend or demonstrate increase in cases in specific time of the year of with environmental conditions. Has this phenomenon observed elsewhere in southeast Asia, in Australia?

5. Also in Fig 2, the positive culture rate seems to be restricted to two monthly periods, April-June and July to October, which correlates with increase in temperature but not with increase precipitation. Any thoughts about these discrepancies?

6. Figure 3 is hard to interpret with clear explanation in the text. What is the meaning of half positive/half negative squares? Further, the color dots describing the ST types are not easily visible and the blue background interferes with visualization. Further, what is the meaning of detecting a ST type in half of the square and not in the other, in squares that are all positive?

7. Lanes 264-265, do the ambient temperature contribute to the growth and persistence? I would argue that ambient temperature contributes to the increase isolation of the bacteria in areas of known prevalence of the pathogen

8. Finally, the discussion is extensive (5.5 pages) and in areas quite confusing. It is highly recommended to streamline the discussion and reduce the length, highlighting the results and comparing with other studies.

9. Lanes 343-344, it is the presence of ST 1994 paddy-specific, or it is a new subtype of concern in Vietnam?.

Minor points:

1. Abstract, lane 29, “…focused on water samples at a…”

2. Lanes 31-32, “B. pseudomallei real-time PCR…gene and also for B. pseudomallei…”.

3. Lane 34, “…quantitative culture method directly…”

4. Lane 56 “…during the summer…”

5. Lane 92, what is “cloud cover”?

6. Lane 119, do you mean: “…August, and autumn…”

7. Lane 124, the following “particularly rice production” is repetitive and the sentence needs to be revised.

8. Lane 164 “…B. pseudomallei RT-PCR…”

9. Lane 184 “(Fig 1A)”

10. Lane 212, what figure panel? Likewise in lane 219 and 222.

11. Lanes 242-244, please revise sentence, particularly in the concept of “complex of multiple factors”

12. Lanes 256-257, “…rainfalls start typically in the middle of…”

13. Lane 271 “…In the environment mimicking…”

14. Lane 280, sentence is redundant: “…an increase to the optimal growth temperature…”

15. Lanes 313-315, this sentence is speculative, and data does not support this statement.

16. References have to be formatted accordingly.

Reviewer #2: A strong study overall in my opinion, with a novel approach that is quite relevant to the epidemiology of melioidosis.

I do have one limitation that I would request be added, with relevant discussion:

Were water temperatures also measured? I am curious whether the drop in the culture rate noted in the August sample may have been due to decreased water temperature secondary to the heavy rains associated with TD1, given that average temperature remained constant from July to October, as an alternative explanation vs dilution mentioned in the manuscript.

Reviewer #3: In summary, this study and its findings have sufficient merits to be published. Many researchers in the B. pseudomallei endemic regions will be interested to read up on it. The main area needing improvement is the method section. It would be of great service to other researchers if this section, in particular the sampling and B. pseudomallei isolation and culture section is meticulously described in detail.

Final recommendation: It may help improve readability further if English language assistance was obtained.

PLOS authors have the option to publish the peer review history of their article (what does this mean? ). If published, this will include your full peer review and any attached files.

**Do you want your identity to be public for this peer review?** For information about this choice, including consent withdrawal, please see our Privacy Policy .

Reviewer #1: No

Reviewer #2: **Yes: ** David D Blaney, MD MPH

Reviewer #3: **Yes: ** Sylvia Daim

**Figure resubmission:**
---

## [Decision Letter · Decision Letter 1]

19 Jun 2025

PNTD-D-25-00046R1Seasonal change of Burkholderia pseudomallei in paddy field water strongly correlates with ambient temperature: A study in north-central VietnamPLOS Neglected Tropical Diseases Dear Dr. Trinh, Thank you for submitting your manuscript to PLOS Neglected Tropical Diseases. After careful consideration, we feel that it has merit but does not fully meet PLOS Neglected Tropical Diseases's publication criteria as it currently stands. Therefore, we invite you to submit a revised version of the manuscript that addresses the points raised during the review process. Please submit your revised manuscript within 30 days Jul 19 2025 11:59PM. If you will need more time than this to complete your revisions, please reply to this message or contact the journal office at plosntds@plos.org. Please include the following items when submitting your revised manuscript:* A rebuttal letter that responds to each point raised by the editor and reviewer(s). You should upload this letter as a separate file labeled 'Response to Reviewers '. This file does not need to include responses to any formatting updates and technical items listed in the 'Journal Requirements' section below.* A marked-up copy of your manuscript that highlights changes made to the original version. You should upload this as a separate file labeled 'Revised Manuscript with Track Changes '.* An unmarked version of your revised paper without tracked changes. You should upload this as a separate file labeled 'Manuscript '. If you would like to make changes to your financial disclosure, competing interests statement, or data availability statement, please make these updates within the submission form at the time of resubmission. Guidelines for resubmitting your figure files are available below the reviewer comments at the end of this letter. We look forward to receiving your revised manuscript. Kind regards, Georgios PappasSection EditorPLOS Neglected Tropical Diseases Georgios PappasSection EditorPLOS Neglected Tropical Diseases

Shaden Kamhawi

co-Editor-in-Chief

Paul Brindley

co-Editor-in-Chief

 **Reviewers' comments:** Reviewer's Responses to Questions

**Key Review Criteria Required for Acceptance?**

**Methods**

-Are the objectives of the study clearly articulated with a clear testable hypothesis stated?

-Is the study design appropriate to address the stated objectives?

-Is the population clearly described and appropriate for the hypothesis being tested?

-Is the sample size sufficient to ensure adequate power to address the hypothesis being tested?

-Were correct statistical analysis used to support conclusions?

-Are there concerns about ethical or regulatory requirements being met?

Reviewer #1: See below

Reviewer #2: Objectives are clear, study design appropriate for the stated objectives. Sample size sufficient, and correct statistical analyses. All issues identified by reviewers appear to have been adequately addressed.

In looking at Figure 1A, consider adding the word "confirmed" for the figure text, as the manuscript refers to only confirmed cases (very minor issue)

**Results**

-Does the analysis presented match the analysis plan?

-Are the results clearly and completely presented?

-Are the figures (Tables, Images) of sufficient quality for clarity?

Reviewer #1: See below

Reviewer #2: Results do match analysis plan and are clearly and completely presented. Figures are sufficient and provide good clarity.

All issues identified by reviewers appear to have been adequately addressed. It is unfortunate that water temperature was not measured at the time of collection, however ambient temperature should be an adequate surrogate. Future studies would be more robust with associated water temperature measures.

**Conclusions**

-Are the conclusions supported by the data presented?

-Are the limitations of analysis clearly described?

-Do the authors discuss how these data can be helpful to advance our understanding of the topic under study?

-Is public health relevance addressed?

Reviewer #1: See below

Reviewer #2: Conclusions provided are well supported, and limitations clearly described. Study does advance understanding of topic, and authors provide suggestions for future study as well as addressing limitations in their study that could be addressed in future studies. Public health relevance is addressed.

Again, all reviewers comments appear to have been adequately addressed.

**Editorial and Data Presentation Modifications?**

Reviewer #1: See below

Reviewer #2: Line 90 - I would suggest one minor word change, from "retains" to "replenishes"

Line 124-125 - Would reword for clarity- I think a word or two is missing (e.g. located between two climate classifications/types

**Summary and General Comments**

Reviewer #1: PNTD-D-25-00046R1

"Seasonal change of Burkholderia pseudomallei in paddy field water strongly correlates

with ambient temperature: A study in north-central Vietnam."

Vu, A.T.N., et al.

This is a significantly improved paper evaluating the presence of Burkholderia pseudomallei in paddy fields in Vietnam and the impact seasonal changes are having in the isolation of the bacteria in the soil. The authors were responsive to the reviewers’ requests, and I only have two minor points to correct:

Minor points:

1. Within the results section, figure 1 panels C, D and E should be discussed and not just described in the figure legend.

2. Similarly, Figure 4 should be better described in the results section and not just in the figure legend.

Reviewer #2: Revisions have substantially improved the quality of the manuscript. I think this study is very important and will lead to further studies that can better describe the myriad of environmental factors influencing B pseudomallei as well as associated melioidosis infections

PLOS authors have the option to publish the peer review history of their article (what does this mean? ). If published, this will include your full peer review and any attached files.

**Do you want your identity to be public for this peer review?** For information about this choice, including consent withdrawal, please see our Privacy Policy .

Reviewer #1: No

Reviewer #2: **Yes: ** David Dwight Blaney

---

## [Editor Report · Decision Letter 2]

7 Jul 2025

Dear Dr Trinh,

We are pleased to inform you that your manuscript 'Seasonal change of Burkholderia pseudomallei in paddy field water strongly correlates with ambient temperature: A study in north-central Vietnam' has been provisionally accepted for publication in PLOS Neglected Tropical Diseases.

Best regards,

Georgios Pappas

Section Editor

Georgios Pappas

Section Editor

Shaden Kamhawi

co-Editor-in-Chief

Paul Brindley

co-Editor-in-Chief

---

## [Editor Report · Acceptance letter]

Dear Dr Trinh,

We are delighted to inform you that your manuscript, "Seasonal change of Burkholderia pseudomallei in paddy field water strongly correlates with ambient temperature: A study in north-central Vietnam," has been formally accepted for publication in PLOS Neglected Tropical Diseases.

Best regards,

Shaden Kamhawi

co-Editor-in-Chief

Paul Brindley

co-Editor-in-Chief
